# Adipose Tissue Dysfunction and Obesity-Related Male Hypogonadism

**DOI:** 10.3390/ijms23158194

**Published:** 2022-07-25

**Authors:** Valentina Annamaria Genchi, Erica Rossi, Celeste Lauriola, Rossella D’Oria, Giuseppe Palma, Anna Borrelli, Cristina Caccioppoli, Francesco Giorgino, Angelo Cignarelli

**Affiliations:** Section of Internal Medicine, Endocrinology, Andrology and Metabolic Diseases, Department of Emergency and Organ Transplantation, University of Bari Aldo Moro, 70124 Bari, Italy; valengenchi@gmail.com (V.A.G.); ericaro1993@gmail.com (E.R.); celeste.lauriola95@gmail.com (C.L.); rossella.doria@uniba.it (R.D.); giuseppe.palma@uniba.it (G.P.); anna.borrelli@uniba.it (A.B.); cristina.caccioppoli@uniba.it (C.C.); francesco.giorgino@uniba.it (F.G.)

**Keywords:** hypogonadism, obesity, adipose tissue, reproductive dysfunction, infertility

## Abstract

Obesity is a chronic illness associated with several metabolic derangements and comorbidities (i.e., insulin resistance, leptin resistance, diabetes, etc.) and often leads to impaired testicular function and male subfertility. Several mechanisms may indeed negatively affect the hypothalamic–pituitary–gonadal health, such as higher testosterone conversion to estradiol by aromatase activity in the adipose tissue, increased ROS production, and the release of several endocrine molecules affecting the hypothalamus–pituitary–testis axis by both direct and indirect mechanisms. In addition, androgen deficiency could further accelerate adipose tissue expansion and therefore exacerbate obesity, which in turn enhances hypogonadism, thus inducing a vicious cycle. Based on these considerations, we propose an overview on the relationship of adipose tissue dysfunction and male hypogonadism, highlighting the main biological pathways involved and the current therapeutic options to counteract this condition.

## 1. Introduction

Obesity, a worldwide health problem, is characterized by excessive fat accumulation. Globally, the prevalence of obesity has nearly tripled since 1975, with about 13% of adults being obese, and it will continue to rise rapidly in the next years. Indeed, one in five adults are predicted to have obesity by 2025 [1]. Obesity is also one of the key risk factors for many noncommunicable diseases, such as coronary heart disease, hypertension and stroke, certain types of cancer, type 2 diabetes, gallbladder disease, dyslipidemia, osteoarthritis and gout, and pulmonary diseases, including sleep apnea, and represents one of the principal causes of secondary hypogonadism in men, a condition characterized by the impairment of the hypothalamic–pituitary–testicular (HPT) axis, in which the reduction in testosterone levels is also accompanied by signs and symptoms of hypogonadism, such as decreases in libido, erectile function, semen quality, strength, and mood [2,3,4,5,6]. The prevalence of hypogonadism in normal-weight adult males is around 32%, while about 75% of subjects with severe obesity (BMI > 40 kg/m^2^) have from hypogonadism [7,8,9,10].

Under physiological conditions, the HPT axis is activated by kisspeptins, peptides able to regulate the secretion of hypothalamic gonadotropin-releasing hormone (GnRH) into the hypothalamic–hypophyseal portal circulation, thus stimulating the release of luteinizing hormone (LH) and follicle-stimulating hormone (FSH) from the pituitary gland. Both hormones travel in the systemic circulation and reach the testes, where LH stimulates the secretion of testosterone from Leydig cells and, along with FSH, supports appropriate spermatogenesis [11]. Obese subjects exhibit decreased levels of total and free testosterone as compared to lean subjects [12], and both BMI and waist circumference are strongly related to the degree of HPT axis dysfunction [13,14]. Furthermore, hypogonadism is associated with fat accumulation, leading to an endless loop in which abnormal adipose tissue (AT) expansion impairs testosterone production, resulting in further accumulation of AT [11].

Several mechanisms may favor hypogonadism; these are mainly related to the excessive amount of AT, particularly in the visceral depot, and to the derangement of its function [15,16]. The impact and contribution of these on the development of obesity-related hypogonadism are not fully elucidated. Herein, we summarize emerging in vivo and in vitro evidence supporting the link between obesity and hypogonadism and the effects of both pharmacological and non-pharmacological strategies to contrast obesity and restore the eugonadal state.

## 2. The Effects of Adipose Tissue Dysfunction on Androgen Levels

### 2.1. Direct Effects of Adipose Tissue

Obesity is a consequence of the increased accumulation of lipids and expansion of AT. As testosterone is a fat-soluble molecule, we can speculate that it is likely to be sequestered into AT depots, leading to reduced circulating levels of this substance in obesity; however, to date, the measurement of steroid concentrations in fat cells and tissue has produced conflicting results [17,18,19]. Indeed, adipocytes from subcutaneous AT (SAT) retrieved from obese men were shown to have higher concentrations of intracellular testosterone as compared with those retrieved from lean men [20]. Despite this greater testosterone accumulation, SAT from obese patients had lower expression of androgen-responsive genes involved in the lipolytic and anti-adipogenic pathways, suggesting the altered function of adipose cells in these individuals. This may, in turn, lead to reduced circulating testosterone levels and reduced fat oxidation in obese men [20].

AT from obese subjects also shows an upregulation of aromatase activity, which is proportional to body fat mass and converts testosterone into estrogens. The increased levels of estrogens, in turn, reduce LH pulse amplitude and may directly upregulate adipogenesis and increase subcutaneous, ectopic, and visceral fat [21,22]. Therefore, the obesity-induced increase in aromatase expression may lead to further peripheral fat accumulation, both by increasing the concentration of estrogens and reducing LH-induced testosterone production [22,23].

Taken together, these results suggest that AT may participate directly in androgen deprivation both by sequestering testosterone from the systemic circulation and through its conversion to estradiol, and indirectly by reducing the hypothalamic stimulus on testosterone synthesis.

### 2.2. Effects of Adipokines

Obesity is also known to alter the release of adipocytokines from AT, resulting in important health consequences [24,25,26]. For instance, the increased secretion of leptin by expanded AT is known to impair central leptin signaling and to participate in the development of metabolic alterations and HPT dysfunction [27,28,29]. Indeed, hyperleptinemia, typical in obesity, is associated with the onset of androgen deficiency and abnormalities of the reproductive tract, sperm count, and motility [15,30,31].

Leptin regulates HPT functionality both through indirect control of forebrain neurons and direct control of testicular tissue. Early experimental studies revealed that kisspeptin neurons are targets of leptin activity. Indeed, leptin-deficient obese male mice show a reduction in kisspeptin production. The levels of kisspeptin were restored, even if only partially, when mice were treated with leptin replacement therapy, and this was associated with an increase in the cellular content of KISS-1 mRNA by 28% [32]. Specifically, both leptin and kisspeptin are co-expressed in neurons of the arcuate nucleus (ARC) of the hypothalamus, the action of which appears to mediate a negative effect on sex steroids and gonadotropin secretion via the control of GnRH pulses at or near GnRH nerve terminals [32,33,34,35] (Figure 1).

Beyond the central regulation of the GnRH pathways, leptin appears to regulate male reproduction by regulating testicular cells. Indeed, leptin can cross the blood–testis barrier, thus modulating the steroidogenic process [36]. In males, the expression of the leptin receptor (LEPR) in the gonads appears to be restricted to Leydig cells; its levels are inversely correlated with testosterone concentration and with abnormalities of sperm quality and production [37]. Leptin acts on testicular tissue principally through the activation of the Janus kinase 2 (JAK2)/signal transducer and activator of transcription 3 (STAT3) pathway [38,39]. Briefly, the activation of JAK2 causes the phosphorylation of several residues of LEPR, which consequently leads to the activation of STAT3, resulting in nuclear translocation and steroid gene transcription (i.e., translocator protein [TSPO], steroidogenic acute regulatory protein [STAR]), or the recruitment of suppressor of cytokine signaling 3 (SOCS3), the key inhibitor of leptin signaling [40,41]. In obesity and hyperleptinemia, negative feedback control of leptin signaling occurs due to the increased expression of SOCS3 inhibiting the phosphorylation of JAK2, thus contributing to the onset of leptin resistance [42] (Figure 1). In this setting, a variation in reproductive parameters occurs, as reported in hyperleptinemic mice, in which reduced in testosterone levels, in association with a decrease in the volume and weight of testes, as well as in the number of spermatocytes, sperm, Leydig cells, and offspring, were noted [43]. All these effects were principally mediated by the upregulation of SOCS3 resulting in the inhibition of JAK2 phosphorylation, showing that this pathway has a pivotal role in the control of androgen levels and reproductive performance [43]. Moreover, Zaho et al. confirmed that both short and prolonged exposure to a high-fat diet (HFD) increases circulating leptin levels in mice together with apoptosis in Leydig cells and reduced sperm count and motility, which are all events correlated with increased oxidative stress in the testes [43,44].

In vitro data also established that supraphysiological levels of leptin inhibited steroidogenic gene expression and the activation of the JAK/STAT pathway in a tumoral line of murine Leydig cells [45]. The same pathway was found to be involved in central regulation of leptin signaling as displayed in mice with neural-specific disruption of STAT3, which recapitulated the phenotype typically observed in leptin resistance (i.e., hyperphagia, diabetes) in association with an increase in infertility [46]. Together, these results indicate that adequate leptin signaling in both testes and neuronal circuits sustains the physiological activity of the HPT axis, and that the presence of leptin-resistance could favor the development of functional hypogonadism and the related reproductive dysfunction.

Among the humoral factors secreted by AT, adiponectin was recently found to regulate the expression of genes implicated in steroidogenesis. Adiponectin release was found to be reduced in the presence of obesity, insulin resistance, and other co-morbidities [47,48,49]. More recently, the downregulation of adiponectin was observed in association with reduced testosterone levels [50]. Both adiponectin and its receptors (AdipoR1 and AdipoR2) are expressed in Leydig cells and seminiferous tubules; thus, the involvement of this hormone in testicular activities is reasonable [51].

Notably, conflicting results have been produced concerning the interplay between adiponectin and the HPT axis. Caminos et al. observed that treatment with adiponectin reduced GnRH-induced testosterone secretion in rat testes [52]. In contrast, when used at high doses, adiponectin stimulated the proliferation and survival of testicular cells, protected them from oxidative stress, and enhanced the transport of energetic substrates [53]. Compelling in vitro evidence also underlined that this adipokine may directly control the function of Leydig cells through the activation of key protein and enzymes involved in androgen synthesis (i.e., STAR, CYP11A1), as well as by promoting testosterone production in a cAMP–PKA-dependent manner [54,55]. Similar results were obtained in diabetic mice with increased body weight and reduced adiponectin levels, with testicular dysfunction improved after adiponectin replacement therapy in terms of upregulation of steroidogenesis, amelioration of insulin receptor-induced transport of energy substrates (glucose, lactate), and a marked reduction in oxidative stress markers in testes [56].

The role of adiponectin in the regulation of testicular function was also clarified by an in vivo model in which the genetic abrogation of AdipoR2 increased adiposity and caused atrophy of the seminiferous tubules and aspermia [57].

Considering these results, we can conclude that adiponectin is essential for the maintenance of the eugonadal state, as well as of normal reproductive activity. Therefore, abnormalities in its production and secretion, as typically observed in the setting of AT dysfunction, may exacerbate obesity-induced hypoandrogenemia.

### 2.3. Effects of Inflammation

Obesity is reportedly accompanied by low-grade inflammation. Under energy storage overload, an increased release of proinflammatory cytokines (i.e., TNF-α, IL-1β, IL-6), especially from the visceral AT, is well documented [58]. Accumulating evidence indicates that the overproduction of inflammatory mediators, besides leading to metabolic complications, is associated with reduced testosterone levels. Several studies have linked various acute and chronic inflammatory conditions (e.g., sepsis, burns, autoimmune diseases) to biochemical testosterone deficiency [59,60,61]. Large-scale epidemiological studies have associated obesity and systemic inflammation (e.g., elevated C-reactive protein (CRP)) with lower testosterone levels [62,63,64], whereas the administration of proinflammatory cytokines or endotoxins to men [65] or the treatment of Leydig cells in culture with proinflammatory cytokines have been shown to decrease testosterone production [66]. Taken together, these data provide strong evidence for a link between inflammation and testosterone deficiency.

Indeed, a negative correlation between IL-6 and testosterone concentrations was found in dysmetabolic men [67]. In a pro-inflammatory environment, a reduction in testosterone levels could occur through modifications of the HPT axis, as demonstrated in cultured anterior pituitary cells from male rats in which IL-6 suppressed GnRH-stimulated LH release [68]. Likewise, IL-1β and TNF-α were found to suppress the release of gonadotropin hormones, particularly LH, by interfering with the translational mechanisms of the GnRH transcript [69,70]. The noxious effect of inflammation on GnRH production is principally due to the reduced responsiveness of hypothalamic neurons to kisspeptins (Figure 1). The role of kisspeptin signaling in the maintenance of the balance of gonadotropins and androgens emerged from previous investigations where the infusion of exogenous intravenous kisspeptin-54 in healthy men stimulated a significant increase in serum LH, FSH, and testosterone concentrations [71]. However, in the presence of a proinflammatory condition, the activity of kisspeptin neurons is compromised, as reported by Sarchielli et al. who demonstrated that prolonged exposure to TNF-α inhibits gene expression of kisspeptin receptors (KISS1R) and affects the cilia volume of hypothalamic cells, thus interfering with the distribution of these receptors on the plasma membrane [72].

In detail, through the activation of TNF-α-induced Ikbeta kinase (IKK-beta)/NF-kB pathway, hypothalamic neurons lose the ability to secrete GnRH and develop a kisspeptin-resistance due to a ciliogenesis defect [72]. Corroborating in vivo results confirmed these data demonstrating the inhibitory effects of TNF-α on the GnRH–LH system and elucidating that C-reactive protein might also well interfere with the HPT axis, as well as participating in metabolic alterations [73].

Interestingly, as described before, several adipokines can directly suppress the HPT axis. However, numerous studies have clearly demonstrated how they can enhance (e.g., leptin or resistin) or alleviate (e.g., adiponectin and ghrelin), various features of inflammatory disease (e.g., asthma), acting either as pro- or anti-inflammatory factors, respectively [74,75]. Therefore, an indirect effect of adipokines mediated by inflammation may be also hypothesized.

### 2.4. Effect of Oxidative Stress

The expansion of AT requires the appropriate formation of new vessels to guarantee an adequate blood supply. Nevertheless, the overfeeding-induced increase in AT might occur at a rate difficult to be paralleled by a sufficient increase in blood flow and oxygen supply, thus leading to AT hypoxia [76]. Hypoxia may underpin oxidative stress in terms of generation of reactive oxygen species (ROS), providing additional routes by which low O_2_ tension can lead to adipose tissue dysfunction [77].

Several stressogenic stimuli are prominent in obesity, including hyperglycemia, hyperleptinemia, and mitochondrial failure, all known contributors of the increased generation of ROS and adipose tissue dysfunction [78,79,80].

Intense oxidative stress may affect the steroidogenic cascade in Leydig cells, eventually leading to decreased testosterone production and infertility. ROS can alter mitochondrial processes in Leydig cells by diminishing the expression of STAR, which can, in turn, decrease the mitochondrial transport of cholesterol and consequently reduce the synthesis of androgens [66,81,82].

In addition, the excessive production of ROS could participate in the onset of functional hypogonadism via an indirect mechanism, by enhancing cortisol secretion, which in turn affects LH secretion from pituitary gland, thus reducing testosterone production by Leydig cells [81] (Figure 1).

In this scenario, the ROS-dependent abrogation of testosterone secretion may also lead to a reduction in nitric oxide (NO) within the corpus cavernosum, thus impairing cavernous relaxation and favoring the development of erectile dysfunction [6,83]. Indeed, several studies have demonstrated that the restoration of the antioxidant system via either pharmacological compounds or genetic methods re-established erectile function through an increase in NO production and appeared to potentiate the efficacy of current pharmacological medications for sexual dysfunction [84,85,86]. Nevertheless, in the tunica albuginea compartment of the corpora cavernosa, ROS have been demonstrated to increase NO as well and its products of peroxidation (e.g., nitric oxide radical, peroxynitrite protonated, peroxynitrite nitrogen dioxide radical, nitronium ion) leading to chronic penile modifications, such as induratio penis plastica [87,88].

## 3. Effects of Pharmacological and Non-Pharmacological Approaches for Obesity on Hypogonadism

The mainstay of treatment for individuals with obesity is lifestyle interventions, such as hypocaloric diets. Nevertheless, notwithstanding their importance, it is undeniable that diet and behavioral therapies have quite an elevated rate of failure in the long-term, as demonstrated by the high rates of obesity relapse. For this reason, different weight-loss drugs have been developed, and several more are currently under investigation; each of them affects different biological and/or neuro-endocrine pathways, reflecting the complexity of energy balance and adipose tissue biology. Finally, bariatric surgery is indicated as the ultimate weight-loss strategy, since it not only reduces the amount of food ingested but also modifies the metabolic profile of the patient.

Many currently approved therapies for weight-loss have been recently investigated for their ability to also ameliorate obesity-induced hypogonadism. Herein, we discuss the current evidence on the effects of pharmacological and non-pharmacological therapies against hypogonadism and obesity on both the HPT axis and AT function.

### 3.1. Pharmacological Treatment

#### 3.1.1. Orlistat

Orlistat, an inhibitor of intestinal lipase, promotes body weight reduction. However, body weight loss obtained with this drug is probably clinically irrelevant, even if statistically significant as compared to placebo (~−3.0 kg) [89]. Therapeutic effects of orlistat on obesity-induced steroidogenesis and spermatogenesis decline have been described in murine models. Indeed, treatment with orlistat has been associated with decreased leptin and increased adiponectin levels, improved sperm parameters, and decreased sperm DNA fragmentation; it also increased the levels of steroidogenic hormones, penile cGMP level, sexual behavior, and fertility outcome in obese male rats fed a high-fat diet [90,91]. However, orlistat treatment has not been proven to induce substantial effects on testosterone levels; thus, the potential recovery from functional hypogonadism in obesity is only speculative.

#### 3.1.2. Glucagon-like Peptide-1 Receptor Agonists

Glucagon-like peptide-1 receptor agonists (GLP-1RA), an anti-diabetic drug class acting as incretin mimetic able to exert pleiotropic effects, represent one of the pharmacological approaches for the treatment of obesity [92,93,94,95].

Recently, the positive effects of GLP-1RA on both male fertility and AT health in obese men have been reported [6,96]. Indeed, studies of the GLP-1RA liraglutide and exenatide have suggested potential benefits when used as a co-therapy for andrological problems. In fact, a retrospective observational study reported that the supplementation of liraglutide to testosterone replacement therapy (TRT) allowed a consistent body weight reduction to be achieved and glycemic targets to be reached, together with a recovery in androgen levels in diabetic obese male subjects with overt hypogonadism [97]. Accordingly, short-term combined therapy with exenatide and metformin, a biguanide drug recommended as a first-line antidiabetic therapy, was found to favor the recovery of normal testosterone levels in association with the correction of sexual dysfunction in obese men with diabetes [98]. A preclinical study also found that liraglutide ameliorates hypogonadism symptoms together with improved glucose and lipid metabolism of AT when administered in orchiectomized rats [99]. Likewise, exenatide-based therapy mitigated the detrimental effects of obesity in HFD-fed male mice in terms of the amelioration of testicular inflammation, but did not yield significant improvements in testosterone levels [100].

### 3.2. Diet

Data from European Male Aging Study provide evidence that testosterone undergoes fluctuations by lifestyle factors according to the grade of weight change. In the setting of modest weight loss (<15%), a slight increase in total testosterone (+2 nmol/L) was observed, probably because of increases in SHBG levels, whereas free testosterone did not change [101]. However, with greater weight loss (>15%), total testosterone was definitely increased (+5.75 nmol/L), with significantly higher levels of free testosterone (+51.78 pmol/L), likely due to the activation of the HPT axis, as evidenced by a significant rise in LH release (+2 U/L) [101]. Nevertheless, the success of diet-induced weight-loss in the recovery of gonadal function is generally influenced by the short period of this intervention as compared with other clinical approaches (i.e., bariatric surgery). Indeed, a meal replacement-based low-calorie diet led to modest weight loss (−9.8%) with moderate increases in testosterone (2.9–5.1 nmol/L) as compared to surgical intervention, which was associated with a more robust weight-lowering effect (−32%) and testosterone restoration (7.8–12.5 nmol/L) [102,103,104,105].

Although previous findings have observed that dietary intervention leads to a weak modification of androgens levels, more recent studies in humans observed that the very low-calorie ketogenic diet (VLCKD) represents an effective tool against obesity and functional hypogonadism [106,107,108,109]. In particular, VLCKD treatment improves glucose homeostasis (83.5  ±  10.6 mg/dL), promotes weight loss (14.9 ± 3.9%), and restores β-cell secretory function in association with the improvement of gonadal functions and testosterone secretion (218.1  ±  53.9%) [107,108]. Therefore, even with the paucity of available data, VLCKD may be considered as an encouraging approach for obesity-related hypogonadism.

### 3.3. Bariatric Surgery

Despite the promising results of dietary intervention, bariatric surgery currently represents the best choice for the treatment of morbid obesity for rapid weight loss and metabolic improvement (i.e., reductions in glycaemia, insulinemia, triglycerides) [110].

Indeed, a recent meta-analysis illustrated that bariatric surgery is more effective in terms of weight-lowering efficacy (−32%) and testosterone recovery (+9 nmol/L) when compared to a low-calorie diet that induced smaller body weight reduction (−9.8%) and only a mild increase in testosterone (+3 nmol/L) [111]. Notably, not only was testosterone production affected by body weight modifications, but estrogen levels were also found to be reduced after bariatric surgery (−23 pmol/L), thus revealing an important adaptation of adipose tissue to weight loss. Importantly, high estrogen levels are known to inhibit kisspeptin expression in hypothalamic nuclei, and thus it is reasonable that the bariatric surgery-induced reduction in estrogens could improve the HPT axis function by upregulating kisspeptin expression in GnRH neurons [112].

Notably, the amelioration of testosterone concentrations after bariatric surgery appears to be correlated with the improvement in different parameters (e.g., body weight, BMI, adiponectin, leptin) regardless of the surgical technique (e.g., biliopancreatic diversion, vertical banded gastroplasty, Roux-en-Y gastric bypass) [113,114,115]. In addition, the loss of visceral AT could have a critical role in the post-surgical restoration of the eugonadal condition. Indeed, the reduction in VAT area assessed by MRI is positively associated with the increase in total testosterone plasma levels after bariatric surgery [116]. In addition to the contribution of weight loss per se, the restoration of normal testosterone levels after bariatric surgery could also be hypothesized to occur via the amelioration of inflammation markers since total testosterone levels were found to correlate significantly with changes in IL-6 and reductions in C-reactive protein [117].

Bariatric surgery appears to improve sexual health in obese men in terms of enhancement of sexual drive and erectile and ejaculatory function proportionately to the weight loss [118,119]. Particularly, after laparoscopic Roux-en-Y gastric bypass surgery, sperm volume and viability increased in association with decreased IL-8 and DNA fragmentation [120]. In contrast, previous findings observed that patients who underwent either gastric bypass surgery or sleeve gastrectomy had a reduction in total sperm count one-year after intervention [120], probably due to a rapid and sizeable excess body weight loss that led to a nutritional deficiency thus worsening the spermatogenesis [121]. Considering these results, surgical interventions produce successful and encouraging outcomes in terms of the patient’s overall health and restoration of eugonadal conditions, but are associated with conflicting data on sperm parameters. Thus, further studies on larger populations are needed to elucidate the effect on reproductive outcomes.

## 4. Effects of Testosterone Replacement Therapy on Adipose Tissue

One of the main therapeutic options for the treatment of functional hypogonadism is represented by testosterone replacement therapy (TRT) [122]. Interestingly, TRT might be an ally in promoting fat mass loss and improving metabolic outcomes through the direct actions of testosterone on AT function [10,123]. Indeed, the exogenous administration of testosterone undecanoate IM 1000 mg every 12 weeks was shown to lead to a significant decrease in body weight (~−15 kg), as well as waist circumference (~−12 cm) [124,125]. A meta-analysis by Corona et al. demonstrated that TRT in men with testosterone deficiency exerts its action mainly on the reduction in fat mass and the increase in lean mass, rather than on overall body weight [111]. Moreover, the loss of fat mass observed after TRT appears to relate particularly to the visceral compartment [106]. Interestingly, in castrated obese mice, testosterone-derived estradiol selectively blocks visceral fat growth by the reduction in adipocyte hypertrophy as well as the generation of new adipose cells [126].

The lipolytic efficacy of testosterone-based therapy is also accompanied by improvements in AT function. Indeed, nonhuman primates with androgen deficiency and dysfunctional white AT (i.e., multilocular and insulin-resistant adipocytes) restored normal AT functions and structure after TRT together with enhanced insulin sensitivity [127]. A recent cross-sectional analysis elucidated the mechanisms behind the anti-adipogenic ability of testosterone, observing that TRT enhanced the expression of genes involved in mitochondrial biogenesis and function (i.e., *nrf1*, *tfam*, *fis1*, etc.), lipid-handling genes (i.e., *dio2*, *prkaca*, and *prkacb*) and browning markers (*pgc1α*) in both visceral AT and preadipocytes from hypogonadal obese men [128].

However, when choosing this therapy, it must be kept in mind that even though TRT represents an appropriate strategy to reduce hypogonadal symptoms and adiposity, exogenous testosterone suppresses pituitary LH secretion and dampens spermatogenesis, exerting negative consequences on fertility [129].

## 5. Conclusions

In conclusion, dysfunctional AT plays a critical role in the development of functional hypogonadism in male obesity via both direct and indirect mechanisms. The dysregulation of adipokine secretion, inflammation, and oxidative stress promoted by fat mass expansion contribute, together with testosterone sequestration and inactivation mediated by AT, to the derangement of the HPT axis. On the other hand, hypogonadism fosters the expansion of AT, typically in the visceral compartment, mainly by lowering lipolytic rates and enhancing adipogenesis (Figure 1).

In this scenario, the reduction in fat mass obtained with pharmacological (i.e., orlistat and GLP-1RA) and non-pharmacological strategies (i.e., hypocaloric diet, VLCKD, bariatric surgery) has been shown to ameliorate testosterone levels and potentially male fertility. A remarkable exception is TRT, for which the metabolic improvement and weight loss attributable to exogenous testosterone is inevitably accompanied by infertility (Table 1).

## Figures and Tables

**Figure 1 ijms-23-08194-f001:**
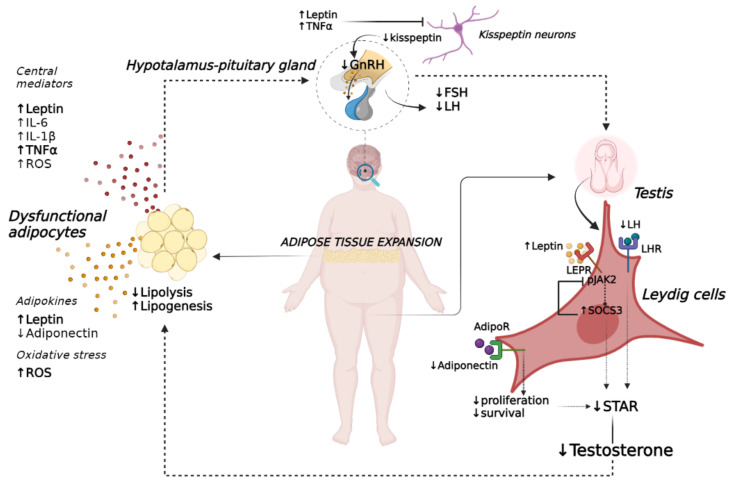
The vicious cycle of obesity–hypogonadism. Obesity-induced AT dysfunction impaired the secretion of several mediators affecting the release of GnRH from the hypothalamus as well as of FSH and LH from the pituitary gland. Low levels of these hormones worsen testicular function in terms of the reduction of testosterone production, leading to overt hypogonadism. The decrease in testosterone levels further exacerbates the impairment of AT in terms of a decreased rate of lipolysis and increased lipogenesis, the extent of which accelerates the failure of HPT axis, thus sustaining a feedforward detrimental loop. Abbreviations: adiponectin receptor (AdipoR); follicle-stimulating hormone (FSH); gonadotropin-releasing hormone (GnRH); interleukin-1β (IL-1β); interleukin-6 (IL-6); suppressor of cytokine signaling 3 (SOCS3); tumor necrosis factor α (TNFα); leptin receptor (LEPR); luteinizing hormone (LH); LH receptor (LHR); phosphorylated Janus kinase 2 (pJAK2); reactive oxygen species (ROS); steroidogenic acute regulatory protein (STAR); signal transducer and activator of transcriptor 3 (STAT3).

**Table 1 ijms-23-08194-t001:** Effects of different approaches for obesity treatment and for hypogonadal hypogonadism on testosterone levels and fertility.

Intervention	Body Weight	T	Fertility
ORLISTAT	↓	-	↑ (?)
GLP-1RA	↓↓	↑	-
VLCKD	↓↓	↑	-
BARIATRIC SURGERY	↓↓↓	↑↑	+/−
TRT	⟷	↑↑	↓

↑ increase; ↓ decrease; - not available; +/− conflicting results; ? uncertain results; ↔ unchanged.

## Data Availability

Not applicable.

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
