# Peer review of "Adipose Tissue Dysfunction and Obesity-Related Male Hypogonadism"

_ijms, 2022, doi:10.3390/ijms23158194_

Round 1

Reviewer 1 Report

General comment

The manuscript entitled “Role of adipose tissue dysfunction in the onset of obesity-related male hypogonadism” aims to summarize and report an updated overview regarding the relationship between adipose tissue and male hypogonadism, with particular attention to testosterone levels and available therapies. Albeit the topic is on point and interesting, the manuscript requires several corrections in order to improve the quality and the readability of the work. In particular, considering the title and the department involved, I would have expected a more molecular-directed article, which was done partly in some paragraphs. In particular, in my opinion, should be expanded the section regarding the physiology of adipose tissue and its relations with testosterone and the hypothalamus-hypophysis axis. Lastly, typos, English grammar and word choice should be revised thoroughly.

The paragraph “Effects of adipokines” is well written and should be used as a benchmark for the others

Corrections in detail are reported below.

-          Major issues

MAIN TEXT

Epidemiological data regarding obesity and hypogonadism should be reported in the introduction, highlighting the future perspective regarding the increasing number of obese subjects during the last years.

73-75: My advice is to improve the explanations regarding the influence of adipose tissue on the hypothalamic-hypophyseal-testicular axis. Reporting main studies would be a nice addition too.

161: It is quite well known that adipose tissue is linked to inflammation. The question is: reduced testosterone levels could be associated with adipose tissue or inflammation independently? Are other inflammatory states associated with hypotestosteronemia and hypogonadism? Data could be collinear.

192-194: It should be said that this is a hypothesis and fewer affirmative sentences should be used.

199-204: Too generic. Report findings of the studies involved.

211: As before, too generic. Avoid redundancy of similar expressions

214-216: This is more complex as reported. If it is true that ROS has several effects on the NO pathway, one of the main effects of ROS on corpora cavernosa is related to the alterations of the tunica albuginea that lead to induratio penis plastica (IPP). To this regard, I would suggest you to see: DOI: 10.26402/jpp.2022.1.05 and DOI: 10.2174/187152812798889321

218: Regarding the association of hypogonadism and obesity (as well as the effects of therapies on testosterone) see also: https://doi.org/10.3390/ijms23073535 and doi: 10.4103/1008-682X.122365

-          Minor issues

ABSTRACT

20-25: I would revise the conclusion of the abstract, reporting that this article proposes an overview regarding the relation of adipose tissue and hypogonadism AND the current therapeutic options, deleting the "discussed as well" as it seems is a secondary aspect as it is written.

MAIN TEXT

28-29: Firstly, references should be placed at the end of the sentences (and this should be revised along with the text); secondly, I would suggest you to remove the italic from secondary hypogonadism, as it would be more suitable for genes.

94-95: report objective data of the cited studies

241: There are not enough references regarding testosterone although.

282: Many data are reported on testosterone and body composition modifications. Expand this section

308: I would add a table which summarized obesity treatments and effects on hypogonadism/testosterone levels

332: The conclusion is redundant with the introduction, especially in the last part

FIGURE

Figure 1 caption is too long. Please summarize and improve the quality of the figure.

Author Response

Reviewer 1

The manuscript entitled “Role of adipose tissue dysfunction in the onset of obesity-related male hypogonadism” aims to summarize and report an updated overview regarding the relationship between adipose tissue and male hypogonadism, with particular attention to testosterone levels and available therapies.

Albeit the topic is on point and interesting, the manuscript requires several corrections in order to improve the quality and the readability of the work.

In particular, considering the title and the department involved, I would have expected a more molecular-directed article, which was done partly in some paragraphs.

In particular, in my opinion, should be expanded the section regarding the physiology of adipose tissue and its relations with testosterone and the hypothalamus-hypophysis axis. Lastly, typos, English grammar and word choice should be revised thoroughly.

The paragraph “Effects of adipokines” is well written and should be used as a benchmark for the others

Corrections in detail are reported below.

Major issues

MAIN TEXT

  • Epidemiological data regarding obesity and hypogonadism should be reported in the introduction, highlighting the future perspective regarding the increasing number of obese subjects during the last years.

As suggested by the Reviewer, we have included data regarding the prevalence of obesity and hypogonadism.

  • 73-75: My advice is to improve the explanations regarding the influence of adipose tissue on the hypothalamic-hypophyseal-testicular axis. Reporting main studies would be a nice addition too.

We appreciate the Reviewer's suggestions; however, we believe that we already considered the role of dysfunctional adipose tissue on the HPT axis in the next paragraphs, including data on the effects of altered secretion of adipokines (i.e., hyperleptinemia) and inflammatory factors on the function of the principal central nuclei (i.e., kisspeptin neurons) regulating GnRH secretion. Indeed, the paragraph entitled “Direct effects of adipose tissue” highlights the current knowledge on the ability of dysfunctional adipose tissue to directly reduce testosterone circulating levels.

  • 161: It is quite well known that adipose tissue is linked to inflammation. The question is: reduced testosterone levels could be associated with adipose tissue or inflammation independently? Are other inflammatory states associated with hypotestosteronemia and hypogonadism? Data could be collinear.

As suggested by the Reviewer, we have clarified the role of the inflammation as a possible independent player affecting testosterone levels negatively.

  • 192-194: It should be said that this is a hypothesis and fewer affirmative sentences should be used.

We thank the Reviewer for the suggestion. We have rephrased the sentence accordingly and inserted a new reference.

  • 199-204: Too generic. Report findings of the studies involved.

We thank the Reviewer for the comment. We reported findings of cited studies in detail.

  • 211: As before, too generic. Avoid redundancy of similar expressions

To comply with the Reviewer's comment, we have expanded this section reporting data related to the effects of oxidative stress on Leydig cells.

  • 214-216: This is more complex as reported. If it is true that ROS has several effects on the NO pathway, one of the main effects of ROS on corpora cavernosa is related to the alterations of the tunica albuginea that lead to induratio penis plastica (IPP). To this regard, I would suggest you to see: DOI: 10.26402/jpp.2022.1.05 and DOI: 10.2174/187152812798889321

We thank the Reviewer for this comment and suggestion. In the revised manuscript, we have now included a more extensive description of the effects of ROS on the NO pathway.

  • 218: Regarding the association of hypogonadism and obesity (as well as the effects of therapies on testosterone) see also: https://doi.org/10.3390/ijms23073535 and doi: 10.4103/1008-682X.122365

To comply with the Reviewer recommendations, we have improved the text in this paragraph by reporting data on the effects of weight-loss induced by diet and bariatric surgery on steroidogenesis. We have further implemented the section of testosterone-replacement therapy by reporting results from different clinical trials, as also suggested.

Minor issues

ABSTRACT

  • 20-25: I would revise the conclusion of the abstract, reporting that this article proposes an overview regarding the relation of adipose tissue and hypogonadism AND the current therapeutic options, deleting the "discussed as well" as it seems is a secondary aspect as it is written.

We thank the Reviewer for this comment. We have modified the abstract as suggested.

MAIN TEXT

  • 28-29: Firstly, references should be placed at the end of the sentences (and this should be revised along with the text).

We thank the Reviewer for this suggestion. We have systematically cited the references at the end of the sentences.

  • Secondly, I would suggest removing the italic from secondary hypogonadism, as it would be more suitable for genes.

We considered the Reviewer's suggestion and have now removed the italic.

  • 94-95: report objective data of the cited studies

To comply with the Reviewer’s comment, we have included detailed information on the effect of leptin on kisspeptin.

  • 241: There are not enough references regarding testosterone though.

We agree with the Reviewer and have now highlighted the hypothetical effect of orlistat on androgens level.

  • 282: Many data are reported on testosterone and body composition modifications. Expand this section

According to the Reviewer’s suggestion, we have improved this section by reporting other clinical studies.

  • 308: I would add a table which summarized obesity treatments and effects on hypogonadism/testosterone levels

To comply with this Reviewer’s suggestion, we have included the main results related to hypogonadism obtained with different approaches for obesity treatment and for testosterone replacement therapy in a new table (Table 1)

  • 332: The conclusion is redundant with the introduction, especially in the last part

We thank the Reviewer for this comment. The text has been modified as suggested.

FIGURE

  • Figure 1 caption is too long. Please summarize and improve the quality of the figure.

We thank the Reviewer for this comment. The text has been modified as suggested.

Reviewer 2 Report

With real interest, I read the manuscript ijms-1789260.

Overall, I find this article interesting and well-written. When I write interesting, I mean it can be interesting not only for researchers in the filed but even for a more general although naturally scientific audience, e.g. of men suffering from obesity, who thanks to this work could realize and understand the problem. So, in my view, a highly valuable contribution.

My comments are minor and/or facultative and are more of suggestions than recommendations.

1.       The Figure summarizing the major messages of the paper is very nice, although I would increase its graphical quality (resolution, etc.).

2.       I would think about additional tables and/or figures illustrating/summarizing some more particular aspects of the story. Especially a table with treatment modalities could be useful.

3.       Lines 84-86. Please, mention also that impaired secretion of leptin (and other adipokines) not only lead to metabolic alterations but also exerts immune/immunomodulatory effects [PMID: 30057383, 34948451].

4.       Whenever you mention the genes, i.e. in line 326, please, write their names in italics.

Author Response

Reviewer 2

With real interest, I read the manuscript ijms-1789260.

Overall, I find this article interesting and well-written. When I write interesting, I mean it can be interesting not only for researchers in the field but even for a more general although naturally scientific audience, e.g., of men suffering from obesity, who thanks to this work could realize and understand the problem. So, in my view, a highly valuable contribution.

My comments are minor and/or facultative and are more of suggestions than recommendations.

  • The Figure summarizing the major messages of the paper is very nice, although I would increase its graphical quality (resolution, etc.).

We thank the Reviewer for this comment. However, we think that there might have been an issue with the submission platform since the original file displays a good resolution image. We will forward this to the editorial manager.

  • I would think about additional tables and/or figures illustrating/summarizing some more particular aspects of the story. A table with treatment modalities could be especially useful.

To comply with the Reviewer comment, we have summarized the main results on the effects of different obesity treatments and testosterone replacement therapy in a new table (Table 1).

  • Lines 84-86. Please, mention also that impaired secretion of leptin (and other adipokines) not only lead to metabolic alterations but also exerts immune/immunomodulatory effects [PMID: 30057383, 34948451].

As properly suggested by the Reviewer, we have included data related to the immunomodulatory effects of adipokines.

  • Whenever you mention the genes, i.e. in line 326, please, write their names in italics.

We thank the Reviewer for this comment. The text has been modified as suggested.

Round 2

Reviewer 1 Report

The authors improve the manuscript accordingly. No further corrections or suggestions are required from my point of view.